# Multipeptide vaccines for melanoma in the adjuvant setting: long-term survival outcomes and post-hoc analysis of a randomized phase II trial

Emily K. Ninmer[1], Hong Zhu [2,3], Kimberly A. Chianese-Bullock [1,3], Margaret von Mehren [4], Naomi B. Haas[4,5], Merrick I. Ross [6], Lynn T. Dengel[1] & Craig L. Slingluff Jr [1,3] ✉

The critical roles of CD4[+] T cells have been understudied for cancer vaccines. Here we report long-term clinical outcomes of a randomized multicenter phase II clinical trial (NCT00118274), where patients with high-risk melanoma received a multipeptide vaccine targeting CD8[+] T cells (12MP) and were randomized to receive either of two vaccines for CD4[+] (helper) T cells: 6MHP (6 melanoma-specific helper peptides), or tet (a nonspecific helper peptide from tetanus toxoid). Cyclophosphamide (Cy) pre-treatment was also assessed. Primary outcomes for T cell responses to 12MP, 6MHP, and tet were previously reported, suggesting immunogenicity of both vaccines but that CD8 T cell responses to 12MP were lower when tet was replaced with 6MHP. Here, in post-hoc analyses, we report durable prolongation of overall survival by adding 6MHP instead of tet. That benefit was experienced only by male patients. A favorable interaction of 6MHP and Cy is also suggested. Multivariable Cox regression analysis of the intent-to-treat population identify vaccine arm (12MP + 6MHP+Cy) and patient sex (male) as the two significant predictors of enhanced survival. These findings support the value of adding cognate T cell help to cancer vaccines and also suggest a need to assess the impact of patient sex on immune therapy outcomes.

Most early cancer vaccines using defined antigens were designed to stimulate cytotoxic CD8[+] T cells with short peptides restricted by Class I MHC molecules. However, CD4[+] helper T cells are crucial for maturation of dendritic cells and for providing cytokines to support CD8[+] T cells[1–3], and they can have direct antitumor activity[4]. Melanoma antigens presented to CD4[+] T cells by Class II MHC were identified later than the Class I associated antigens, and even now,

algorithms for predicting epitopes for CD4[+] T cells lag behind those for CD8[+] T cells. In our early trials, we primarily targeted CD8[+] T cells, but included a tetanus toxoid peptide restricted by Class II MHC that was effective for activating CD4[+] T cells but that was not cognate for melanoma[5]. Vaccination with a cocktail of 12 melanoma peptides restricted by Class I MHC molecules (12MP, NSC#728925) plus a tetanus toxoid helper peptide (tet, NSC#728927) induced CD8[+] T

[1]Department of Surgery/Division of Surgical Oncology and the Human Immune Therapy Center, Cancer Center, University of Virginia, Charlottesville, VA, USA. [2]Department of Public Health Sciences, University of Virginia, Charlottesville, VA, USA. [3]University of Virginia, School of Medicine, Cancer Center, Charlottesville, VA, USA. [4]Fox Chase Cancer Center, Philadelphia, PA, USA. [5]University of Pennsylvania, Philadelphia, PA, USA. [6]Department of Surgical Oncology, MD Anderson Cancer Center, Houston, TX, USA. ✉e-mail: cls8h@uvahealth.org

cell responses to the 12MP in 100% of patients[6]. As melanoma helper peptides were identified, we developed a vaccine comprising 6 melanoma peptides presented by Class II HLA-DR molecules (6 melanoma helper peptides, 6MHP, NSC#728926), which induced Th1-dominant CD4[+] T cell responses in most patients and induced durable clinical activity in some[7,8]. Also, overall survival of stage IV patients vaccinated with 6MHP significantly exceeded that of a matched cohort of unvaccinated patients[9].

Thus, we designed a multicenter, randomized phase II trial to test whether vaccinating with 12MP + 6MHP would enhance CD8[+] T cell responses to 12MP, and would enhance clinical outcomes (NCT00118274, Mel44)[10]. In addition, pretreatment with low-dose cyclophosphamide (Cy) was tested in this study based on prior work showing reduction of regulatory T cells and improved T cell responses and/or tumor control when combined with cancer vaccines in murine models and in non-randomized clinical trials[11–16]. The trial design is represented in Fig. 1A. In our initial report of this trial, we found that 6MHP induced circulating T cell responses to 6MHP in most participants, detected ex vivo; however, CD8[+] T cell response to 12MP was decreased by adding 6MHP instead of tet, in contrast to our hypothesis[10]. Short follow-up at the time of that early report limited

assessments of clinical outcomes[10]. We continued to follow these participants for survival and recurrence annually. We are now 15 years from the enrollment of the last participant on this trial and have evaluated long-term clinical outcomes. For this analysis, we hypothesized that participants vaccinated with 12MP + 6MHP would experience longer overall survival (OS) and recurrence-free survival (RFS) than those vaccinated with 12MP + tet. Thus, long-term outcomes were analyzed by arm and by grouping arms based on helper peptide regimen and cyclophosphamide regimen (Fig. 1). We also hypothesized that there may be differences in clinical outcome as a function of age, stage, or biologic sex.

In this work, long-term follow-up data reveal durable prolongation of OS by adding cognate helper peptides to a vaccine otherwise designed to stimulate CD8[+] T cells. These data also provide support for clinical benefit of targeting non-mutated shared antigens. Benefit appears confined to males. Possible benefit of Cy pretreatment is also suggested, when combined with cognate T cell help.

## Results

Follow-up data were collected for all 167 eligible participants (intention-to-treat (ITT) population), including 82 vaccinated with

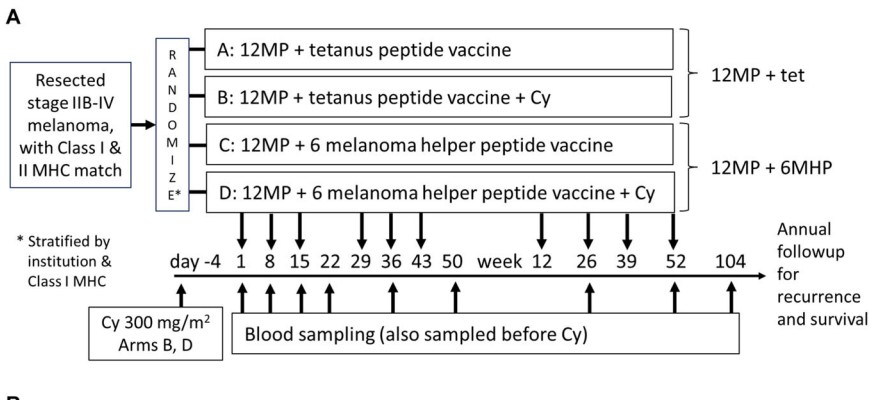

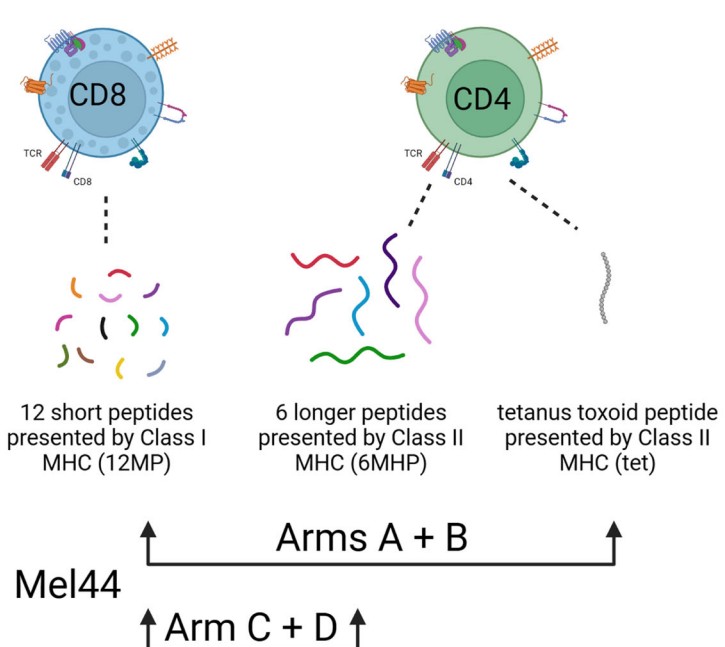

**Fig. 1 | Mel44 clinical trial design and long-term outcomes analysis.** The Mel44 clinical trial was designed to evaluate immunologic and clinical outcomes after vaccination with either of two peptide vaccine formulations, and with or without a dose of cyclophosphamide pretreatment. **A** Protocol schema (**B**) 2 × 2 design to perform comparisons of Arms A and B (12 class I peptides (12MP) plus tetanus helper peptide) to Arms C and D (12 class I peptides (12MP) plus 6 melanoma helper peptides (6MHP)), as well as comparisons of Arms A + C (no Cy) to Arms B + D (+Cy). Panel **B** was created with BioRender.com.

12MP+tet (Arms A + B) and 85 vaccinated with 12MP + 6MHP (Arms C + D). The median follow-up interval was 8.6 years for all participants and 12.2 years for participants alive at the date of last known follow-up, as of data lock in April 2023. The vaccine groups were well-matched by demographic and clinical characteristics (Table 1). For each group, median age was between 55 and 60 years, approximately two-thirds of the participants were male, and ~80% of participants had stage II-III disease.

For the original clinical trial protocol, the primary outcomes were safety and CD8 T cell response to 12MP, which have been reported[10], while a secondary endpoint was disease-free survival (DFS). For the present ad hoc analyses of long-term followup, overall survival (OS) and recurrence-free survival (RFS) have been analyzed by treatment regimen, and subgroup analyses have explored potential impact of age, sex, and stage.

### Survival outcomes for eligible intent-to-treat population by vaccine regimen

Kaplan–Meier survival estimates for the ITT population by vaccine regimen (Arms A + B vs. C + D) are shown in Fig. 2. After 2.5 years from enrollment, the OS estimates diverged progressively to favor the 6MHP vaccine (HR 0.65, 95% CI: 0.40 to 1.05, $p = 0.08$; Fig. 2A). Median OS intervals for 12MP + tet and 12MP + 6MHP were 12.9 years and not reached, respectively. For 12MP + 6MHP, OS rate estimates (±standard error, SE) at 5, 10, and 15 years were 74% (±5%), 68% (±5%), and 61% (±6%), respectively (Fig. 2A). which were numerically lower for 12MP + tet: 68% (±5%), 56% (±6%), and 45% (±7%), respectively (Fig. 2A). After 8 years, the OS estimates for the 12MP + 6MHP vaccine exceeded the upper bound of the 95% confidence interval for 12MP+tet vaccine (Fig. 2B), suggesting durable enhancement of long-term survival by addition of the cognate helper peptides, 6MHP.

Assignment of hazard ratios comparing two survival curves is based on an assumption of proportion hazards over time, but immunotherapy can provide delayed benefit, which challenges the proportional hazard assumption and compromises comparison of outcomes over time[17,18]. An approach recommended to manage this scenario is to report the HR for the curves after they separate. Thus, we performed a landmark analysis for OS by vaccine regimen at 2.5 years (Fig. 2C), which supports enhanced OS with 6MHP vaccination (HR 0.52, 95% CI: 0.27–0.97, $p = 0.04$).

Recurrences were mostly metastases but also included two new primary melanomas (one each, Arms A and C) and one participant with new skin lesions for which records are unclear if they are new primaries or metastases. Participants vaccinated with 12MP+tet and 12MP + 6 MHP had median RFS of 2.7 years and 13.3 years, respectively, weakly favoring the 6MHP vaccine (HR 0.77, 95% CI: 0.51 to 1.18, $p > 0.22$, Fig. 2D). It is notable that there were very few recurrences in either arm after 4 years, with levelling of RFS curves thereafter, continuing to 15 years (Fig. 2D).

### Survival outcomes for eligible intent-to-treat population by cyclophosphamide pretreatment

OS and RFS were also assessed as a function of Cy pretreatment (Arms A + C vs Arms B + D). For OS, the curves are very similar (HR = 0.88, 95% CI 0.55 to 1.42, Supplementary Fig. 1A). For RFS, there is a weak trend favoring the Cy-pretreated participants (HR 0.79, 95% CI, 0.52 to 1.19, Supplementary Fig. 1B).

### Survival outcomes for eligible intent-to-treat population across all four study arms

For the individual study arms A-D, the most favorable OS and RFS outcomes were for those on Arm D, treated with 12MP + 6MHP plus Cy, and the least favorable were for those on Arm A, treated with 12MP+tet

**Table 1 | Participant characteristics by vaccine regimen**

|  | Arms A + B (12MP + tet) (n = 82) | Arms C + D (12MP + 6MHP) (n = 85) | Total (n = 167) | p value[a] |
|---|---|---|---|---|
| Male, n (%) | 53 (65%) | 59 (69%) | 112 (67%) | 0.51 |
| Institution, n (%) |  |  |  |  |
| Fox Chase Cancer Center | 15 (18%) | 13 (15%) | 28 (17%) | 0.87 |
| MD Anderson Cancer Center | 25 (30%) | 27 (32%) | 52 (31%) |  |
| University of Virginia | 42 (51%) | 45 (53%) | 85 (52%) |  |
| Diagnosis at enrollment, n (%) |  |  |  |  |
| Initial diagnosis | 37 (44%) | 48 (55%) | 85 (50%) | 0.14 |
| Recurrence | 45 (56%) | 37 (45%) | 82 (50%) |  |
| AJCC v8 stage at enrollment, n (%) |  |  |  |  |
| IIB-C | 9 (11%) | 15 (18%) | 24 (14%) | 0.37 |
| IIIA | 10 (12%) | 10 (12%) | 20 (12%) |  |
| IIIB-D | 44 (54%) | 44 (53%) | 88 (53%) |  |
| IV (M1a/b) | 12 (15%) | 14 (16%) | 26 (16%) |  |
| IV (M1c/d) | 7 (9%) | 2 (2%) | 9 (5%) |  |
| ECOG PS score 0, n (%) | 73 (89%) | 78 (92%) | 151 (90%) | 0.55 |
| LDH level at enrollment |  |  |  |  |
| N evaluable | 81 | 85 | 166 |  |
| N < ULN (%) | 77 (95%) | 83 (98%) | 160 (96%) | 0.43 |
| N < 1.5× ULN (%) | 81 (100%) | 85 (100%) | 166 (100%) | -- |
| Median age, years (range) | 59.7 (28.8–81.7) | 56.7 (21.4–77.3) | 58.2 (21.4–81.7) | 0.29 |
| Class I MHC allele, n (%) |  |  |  |  |
| HLA-A1 | 25 (30%) | 28 (33%) | 53 (32%) | 0.73 |
| HLA-A2 | 44 (54%) | 41 (48%) | 85 (51%) | 0.48 |
| HLA-A3/31 | 37 (45%) | 41 (48%) | 78 (47%) | 0.69 |

*12MP* 12 class I MHC-restricted melanoma peptides, *tet* tetanus toxoid helper peptide, *6MHP* mixture of six melanoma-specific helper peptides, *AJCC* American Joint Committee on Cancer, *ECOG PS* Eastern Cooperative Oncology Group Performance Status, *LDH* lactate dehydrogenase, *ULN* upper limit of normal, *MHC* major histocompatibility complex, *HLA* human leukocyte antigen.
[a]Chi-square or Fisher's exact test (two-sided) for categorical variables; Mann–Whitney test (two-sided) for age.

without Cy (Supplementary Fig. 2). Comparing each pair of arms, the greatest difference was between Arms A and D, with HR 0.56 (95% CI, 0.28–1.11) favoring Arm D for OS (Supplementary Fig. 2A), and HR 0.61 (95% CI, 0.33–1.10) for RFS (Supplementary Fig. 2B). The power to compare among all 4 arms is limited by the smaller number of participants in each arm.

A landmark analysis at 2.5 years for OS identifies the most favorable outcomes for Arms D vs A (HR 0.40, 95% CI 0.16–0.99) and for Arms D vs B (HR 0.38, 95% CI 0.16–0.91, Supplementary Fig. 2C). Thus, these data leave open the possibility of a favorable OS impact of adding 6MHP and Cy pretreatment.

### Subgroup analysis of associations of age with outcomes

There were significant differences in OS by age subgrouped by decade, specifically better for those ≤40 years, though outcomes among other age groups were similar to each other ($p < 0.002$; Supplementary Fig. 3A). There were no significant differences in RFS by age group, though participants age 40 or less also trended to better RFS (Supplementary Fig. 3B). Within each vaccine regimen, OS was better for age ≤40 Supplementary Fig. 3C), but RFS did not differ by age (Supplementary Fig. 3D).

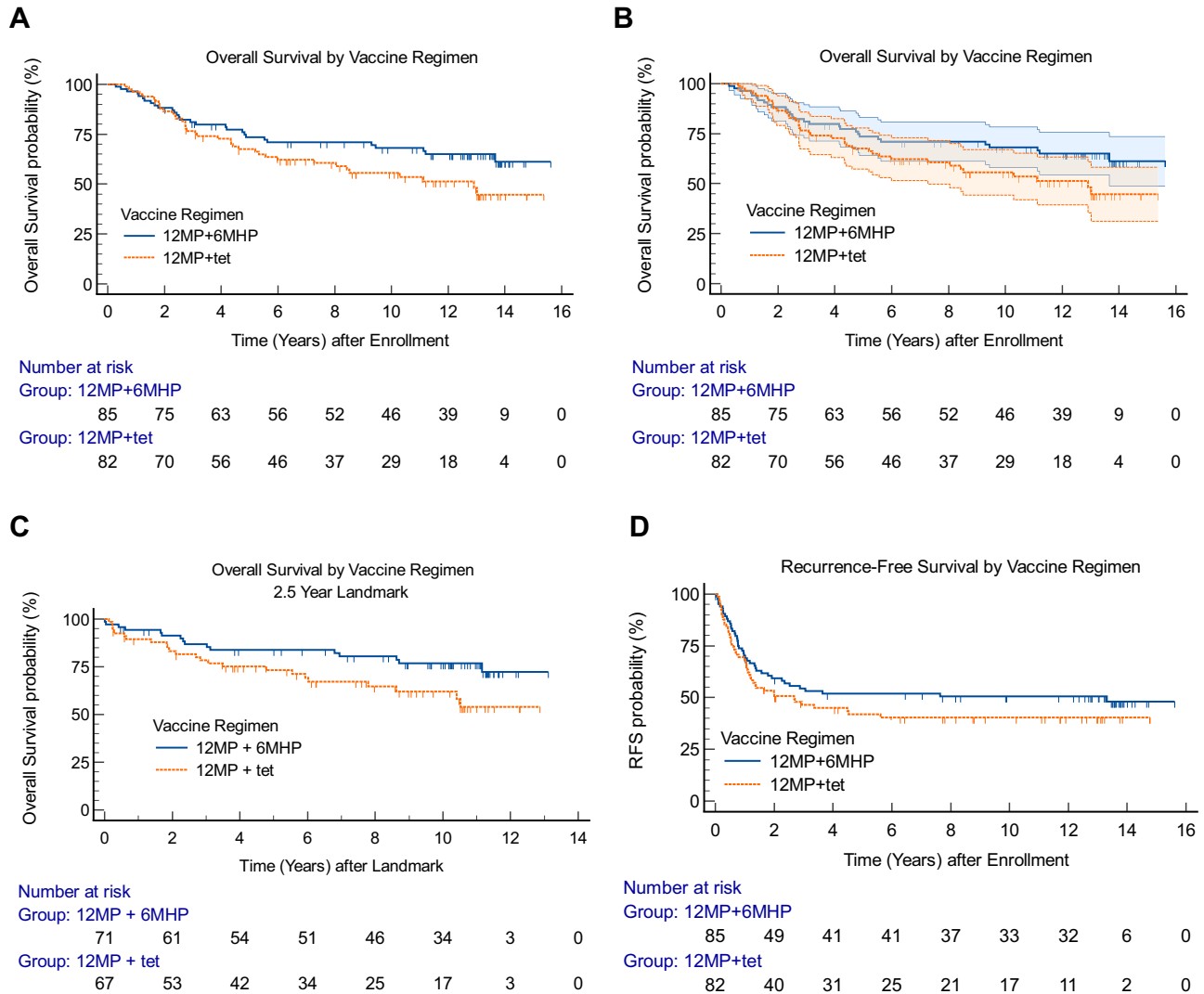

**Fig. 2 | Overall survival and recurrence-free survival by vaccine regimen.**
**A** Kaplan–Meier curves for OS for all eligible participants ($n = 167$) by vaccine regimen (HR 0.65, $p = 0.08$) and **B** with 95% CI shown in shaded regions, **C** Landmark analysis for OS at 2.5 years for eligible participants ($n = 138$) by vaccine regimen (HR 0.52, $p = 0.04$), **D** RFS for all eligible participants ($n = 167$) by vaccine regimen (HR 0.77, $p > 0.22$). $P$ values are from two-tailed logrank tests. Adjustments were not made for multiple comparisons. Source data are provided as a Source Data file.

### Subgroup analysis by biologic sex

Biologic sex of the participants was identified in the case report forms for all participants. Most participants were males (67%), and were well-matched by vaccine regimen (Supplementary Table 2). Kaplan–Meier estimates revealed a trend to improved OS for 12MP + 6MHP among males ($p = 0.08$), but not among females (Fig. 3A). A similar trend for RFS favoring the 6MHP vaccine was found in males, with early separation of the curves ($n = 112$, HR 0.59, 95% CI 0.34–1.02, $p = 0.06$), but there was no benefit for females ($n = 55$, HR 1.29, 95% CI 0.67–2.49) (Fig. 3B). Since the vaccine regimen continued for 1 year, recurrences during the vaccine regimen may reflect suboptimal vaccine-induced immune protection. Thus, we also evaluated RFS for males after a landmark at 1 year: this revealed significant prolongation of RFS for males who received the 6MHP vaccine (HR 0.35, 95% CI 0.14–0.86, $p = 0.02$; Fig. 3C).

### Subgroup analysis by AJCC stage

Patients with earlier stage melanoma (AJCC v8 stage IIB-III) represented 79% of all eligible participants (Table 1) and were well-matched across the vaccine regimens (Supplementary Table 3). Their OS was improved with 6MHP vaccines vs. tet (HR 0.57, 95% CI 0.33–0.97,

$p = 0.037$; Fig. 4A). A favorable impact on RFS was also identified for stage IIB-III patients with 6MHP vaccination (HR 0.61, 95% CI 0.38–0.99, $p = 0.045$; Fig. 4B). Among stage IV patients, neither OS nor RFS differed demonstrably by vaccine regimen (Fig. 4C, D) but the small size of that subset limits the assessment of impact.

### Cox regression analyses

Participants among the 4 study arms were well-matched, but to account for impact of covariates on outcome, we performed multi-variable Cox regression analyses to assess the impact of study arm (A, B, C, D), vaccine regimen (AB vs CD), or Cy arms (AC vs BD), controlling for age (≤40 vs >40), sex, AJCC stage (v8 stage II-III vs IV), study arm, advanced disease status (initial diagnosis vs recurrence), ECOG performance status (0 vs 1), and lactate dehydrogenase (LDH) level (elevated (6) vs normal (160) or not done (1)).

For OS, these variables defined a significant model with 3 significant covariates: study arm, sex, and age. When covariates with $p$ values > 0.1 were removed, the final model included the same 3 significant covariates: age (≤40 vs >40 years, HR 0.049, $p = 0.003$), study arm (A vs D, HR 2.362, $p = 0.018$) and sex (female vs male, HR 1.755, $p = 0.032$; Table 2).

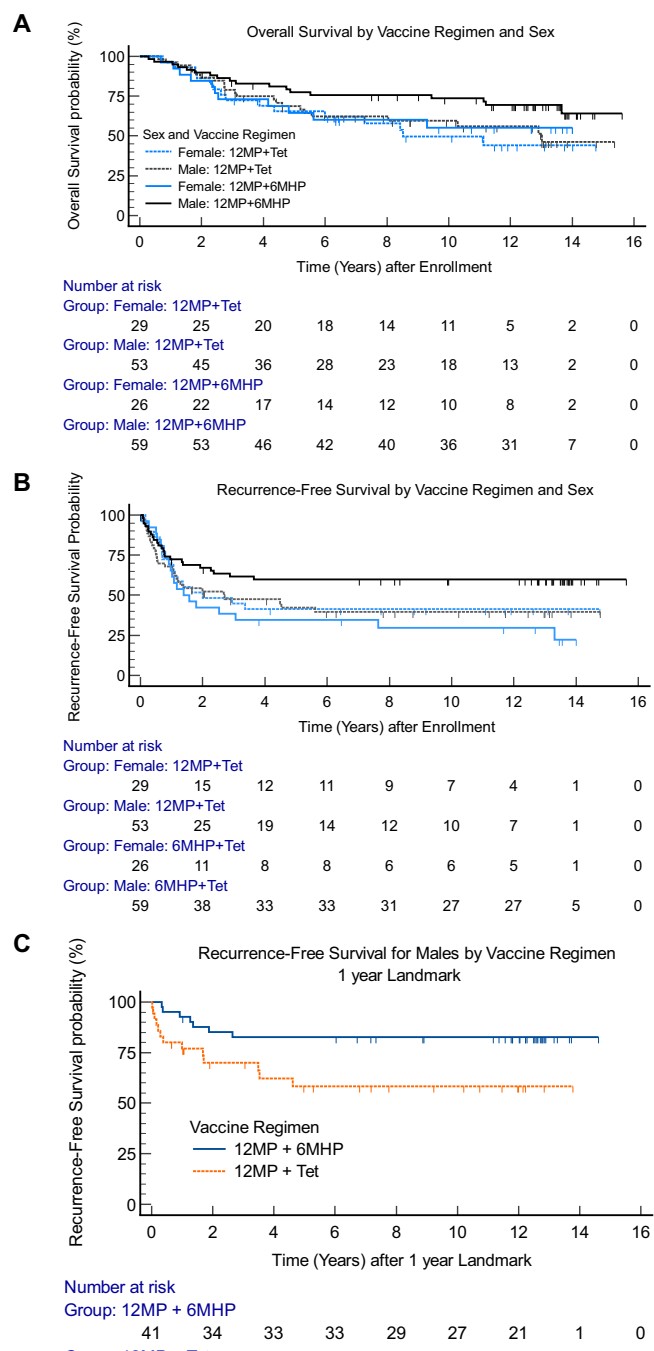

**Fig. 3 | Overall survival and recurrence-free survival by vaccine regimen and sex. A** OS for vaccine regimen and sex (*p* = 0.185 overall): for males/6MHP vs males/tet, HR 0.59 [95% CI 0.33–1.05]; vs female/tet, HR 0.50 [95% CI 0.25–1.01]; vs. females/6MHP, HR 0.61 [0.29–1.26], **B** RFS for vaccine regimen and sex (*p* = 0.075 overall): for males/6MHP vs males/tet, HR 0.58 [95% CI 0.35–0.97]; vs female/tet, HR 0.61 [95% CI 0.34–1.12]; vs. females/6MHP, HR 0.47 [0.25–0.89], **C** Landmark analysis for RFS at 1 year for males (*n* = 77) by vaccine regimen (HR 0.35, *p* = 0.02). *P* values are from two-tailed logrank tests. Adjustments were not made for multiple comparisons. Source data are provided as a Source Data file.

In that analysis, vaccine regimen and Cy arms were not significant when study arm was included as a covariate. Thus, we also performed multivariable analysis with the vaccine regimen, Cy arms, and the same 6 clinical covariates, but without study arm. An optimized model was similar with significantly better survival with younger age (*p* = 0.003),

male sex (*p* = 0.034), and AJCC stage II-III (*p* = 0.047), and with trends to better survival with regimens containing 6MHP (*p* = 0.082) and Cy (*p* = 0.083, Supplementary Table 4).

An optimized model for RFS (*p* = 0.006) was driven by age (*p* = 0.014), sex (*p* = 0.034) and study arm (*p* = 0.036; Supplementary Table 5). We also ran multivariable analysis for RFS excluding arm, and no covariates were significant, but age, sex, and advanced disease status approached significance (Supplementary Table 6).

Among participants evaluable after the 2.5 year landmark (*n* = 138), Cox regression analysis identified significant associations with OS for study arm (A vs D, *p* = 0.019), age and ECOG PS, if study arm was input in the model (Table 3). If vaccine regimen and Cy arms were input instead of arm, the refined model included vaccine regimen (*p* = 0.017), age and ECOG PS; Table 3). Thus, in both univariate and multivariable analyses, cognate CD4 help (6MHP) was a significant driver of OS. Sex was not a significant covariate for OS in the Cox regression analysis for this 2.5 year landmark dataset, but the HR for OS by vaccine regimen was more favorable for males (0.49; 95% CI 0.23–1.04) than for females (0.62; 95% CI 0.20–1.94), favoring the 6MHP regimen. Also, OS estimates at 10 years (7.5 years from the landmark) were 0.84 ± 0.05 (SE) and 0.72 ± 0.11 for males and females, respectively, receiving 6MHP vaccines and 0.70 ± 0.07 and 0.63 ± 0.11 for those receiving tet vaccines (Supplementary Fig. 4).

Models for RFS were also assessed in landmark analyses. Only 83 participants were evaluable for RFS after 2.5 years, and no model with all the variables has significance overall. Instead, we developed a multivariable model for RFS at a 1 year landmark (*n* = 114), where sex was significantly associated with RFS (*p* = 0.038), with HR 2.029 for females vs males. However, no other covariates were significant (Supplementary Table 7).

A provocative finding in univariate analyses was the association of sex with outcome, but only for participants treated with 12MP + 6MHP (Fig. 3A, B). To assess the significance of these findings further, we performed multivariable Cox regression modeling within the 85 participants on Arms C + D. Among the covariates tested (age, study arm, AJCC stage, sex, advanced disease status, ECOG performance status, LDH level), only sex and age were significantly associated with OS (Supplementary Table 8). Modeling the covariates for RFS also identified sex as the only significant association (female vs male, HR 2.308, *p* = 0.016; Supplementary Table 9). That model for RFS for Arms C + D was refined stepwise by excluding covariates with *p* > 0.1, leaving a model with just sex (*p* = 0.004) and age (≤40 vs >40, *p* = 0.04; Supplementary Table 9). On the other hand, Cox regression modeling did not identify an impact of sex on OS (*p* = 0.643, Supplementary Table 10) or RFS (*p* = 0.842, Supplementary Table 11) among participants on arms A + B, when age, AJCC stage, advanced disease status, LDH level, or ECOG status were included in the model.

Conversely, Cox regression modeling for OS for male participants identified only two significant covariates: age (*p* = 0.043) and vaccine regimen (12MP + tet vs 12MP + 6MHP, HR 1.964, *p* = 0.044, Supplementary Table 12). Regression modeling for OS for female participants led to a significant model when inputting vaccine arm, age groups, advanced disease state, and AJCC stage, where age was the only significant covariate, and there was only a weak trend for better outcome with Arm D vs Arm A (Supplementary Table 13). We did explore if there is an interaction between sex and vaccine regimen, by adding an interaction term (Sex*Vaccine regimen), to the models for OS and RFS for the ITT population, and for OS for the 2.5 year landmark population: those *p* values for the interaction term were 0.165, 0.081 and 0.207, respectively.

To understand factors that may explain findings in univariate analyses within stage II-III participants, we performed multivariable analysis including the same covariates as for the ITT population, except stage, and after stepwise refinement, the only significant covariates were vaccine regimen (*p* = 0.035 for OS, 0.045 for RFS), and age

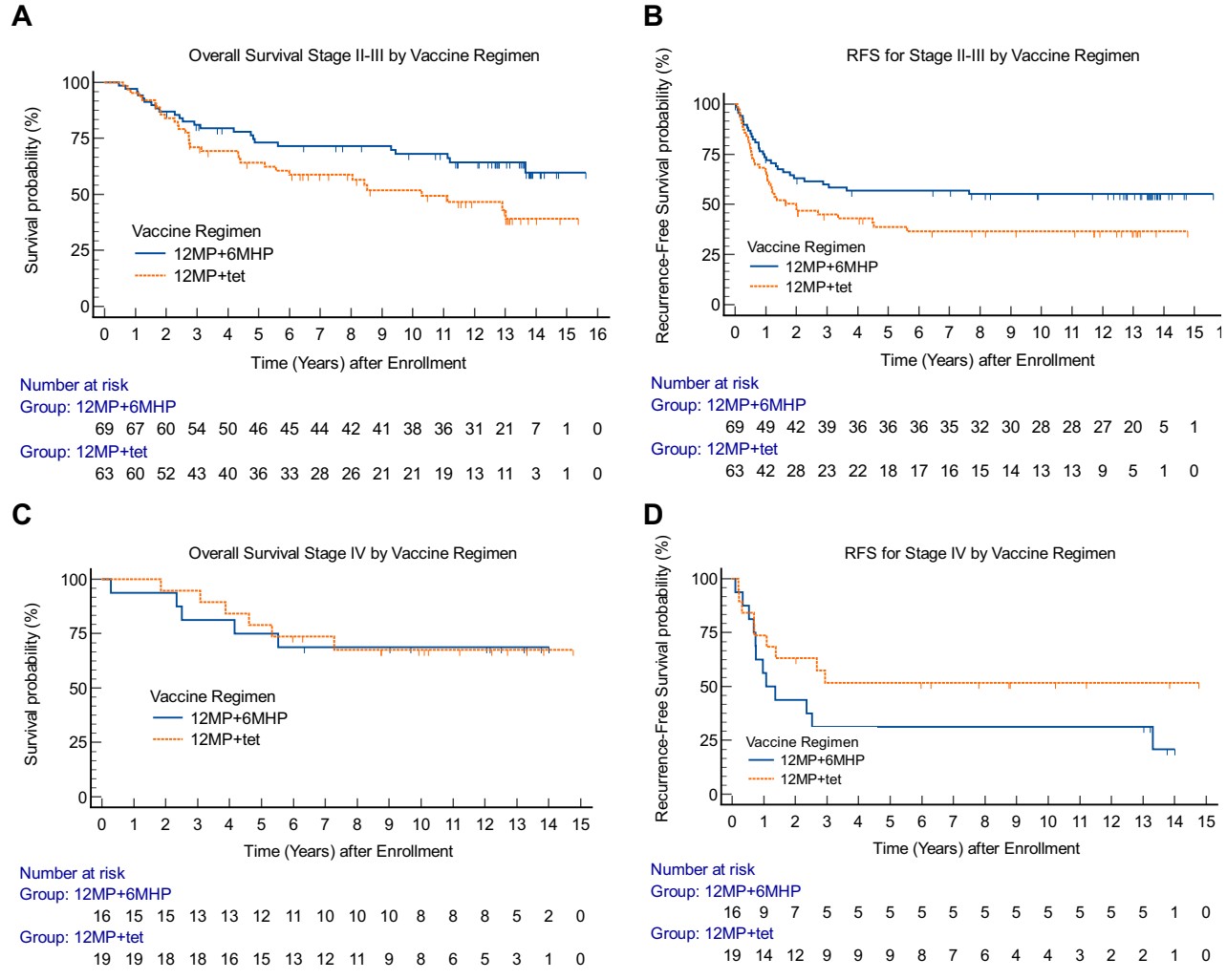

**Fig. 4 | Overall survival and recurrence-free survival by vaccine regimen and AJCC v8 stage. A** OS by vaccine regimen for stage II/III participants (HR 0.57, $p = 0.037$), **B** RFS by vaccine regimen for stage II-III participants (HR 0.61, $p = 0.045$), **C** OS by vaccine regimen for stage IV participants ($p = 0.94$), **D** RFS by vaccine regimen for stage IV participants ($p = 0.17$). *P* values are from two-tailed logrank tests. Adjustments were not made for multiple comparisons. Source data are provided as a Source Data file.

($p = 0.0067$ for OS and 0.067 for RFS; Supplementary Table 14). There were only 35 patients with stage IV melanoma; so, there were too few to include all the variables in a model.

Outcome differences for Cy pretreatment. Cy pretreatment did not impact OS either in the ITT population (Supplementary Fig. 1A) or for the male participant subset (Supplementary Fig. 5A). However, there was a trend to prolonged RFS in the ITT population (Supplementary Fig. 1B) and for the male participants (Supplementary Fig. 5B). Interestingly, among males, the best and worst RFS data are for Arms D and A, respectively, with a HR of 0.42 (95% CI 0.20–0.89, Supplementary Fig. 6), leaving open the possibility that Cy may delay recurrence most when administered with a vaccine that includes cognate T cell help, in this population. Similarly, when modeling outcomes with Cox regression, best OS was observed in Arm D and worst OS in Arm A, suggesting that addition of Cy pretreatment to cognate CD4 help (6MHP vaccine) may be most favorable (Table 2).

## Discussion
This manuscript reports long-term outcomes for participants with high-risk melanoma on a randomized multicenter clinical trial (Mel44, NCT00118274) that tested two experimental multipeptide melanoma

vaccines with or without Cy pretreatment. Participants received either of two vaccine strategies, each of which included a cocktail of short peptide antigens for CD8$^+$ T cells (12MP): in addition, participants received either a tetanus toxoid peptide that stimulates CD4$^+$ T cell responses (tet)[5] or a cocktail of 6 melanoma-specific peptides antigens for CD4$^+$ helper T cells (6MHP)[7]. OS estimates for the two vaccine regimens were similar early, then they separated progressively, favoring the 12MP + 6MHP vaccine regimen. After 8 years, the OS estimates for 12MP + 6MHP participants exceeded the upper bound of the 95% CI for the 12MP + tet participants, evidence that the melanoma-cognate peptides (6MHP) significantly improved long-term OS. Interestingly, the benefit of adding 6MHP was experienced disproportionately by males. Cy pretreatment did not induce significant benefits across the whole study, but multivariable analyses identified the best outcomes for participants on Arm D and the least favorable outcomes for those on Arm A, raising the possibility of a favorable interaction between Cy pretreatment and vaccination with 12MP + 6MHP.

This phase 2 trial was not powered for comparison of OS or RFS; so, even a promising trend for enhanced OS or RFS between the two vaccine regimens is at least hypothesis-generating and could justify a future definitive trial powered to test clinical benefit. A traditional log-

**Table 2 | Cox regression model for overall survival for intention-to-treat population (n = 167)**

| Covariate | Detail | p value | HR | 95% CI |
|---|---|---|---|---|
| **Initial model (p < 0.001, Chi-squared 36.7), OS all eligible patients (n = 167) including study arm and 6 clinical covariates** | | | | |
| Age | ≤40 vs >40 years | **0.002*** | 0.045 | 0.006 to 0.335 |
| Sex | Female vs male | **0.023˙** | 1.844 | 1.089 to 3.123 |
| Study Arm | Arm A vs D | **0.015˙** | 2.444 | 1.189 to 5.022 |
| | Arm B vs D | 0.328 | 1.442 | 0.693 to 3.003 |
| | Arm C vs D | 0.344 | 1.429 | 0.623 to 2.990 |
| Advanced disease status | Recurrence vs initial diagnosis | 0.241 | 0.733 | 0.437 to 1.232 |
| LDH level | High vs normal or not done | 0.219 | 2.153 | 0.634 to 7.311 |
| AJCC stage, v8 | Stage IV vs II-III | 0.198 | 0.631 | 0.314 to 1.271 |
| ECOG PS score | 1 vs 0 | 0.439 | 1.329 | 0.647 to 2.733 |
| **Refined model (p < 0.0001, Chi-squared 33.4) OS all eligible patients (n = 167) Beginning with study arm, vaccine regimen and Cy arms, with stepwise refinement** | | | | |
| Age | ≤40 vs >40 years | **0.003*** | 0.049 | 0.007 to 0.354 |
| Sex | Female vs male | **0.032˙** | 1.755 | 1.050 to 2.932 |
| AJCC stage, v8 | Stage IV vs II-III | **0.048*** | 0.520 | 0.272 to 0.995 |
| Study Arm | Arm A (12MP+tet) vs D (12MP + 6MHP+Cy) | **0.018˙** | 2.362 | 1.159 to 4.812 |
| | Arm B vs D | 0.332 | 1.431 | 0.694 to 2.947 |
| | Arm C vs D | 0.345 | 1.425 | 0.684 to 2.970 |

*HR* hazard ratio, *CI* confidence interval, *LDH* lactate dehydrogenase, *AJCC* American
Joint Committee on Cancer, *ECOG PS* Eastern Cooperative Oncology Group Performance Status.
˙Significant, *p* < 0.05 (bolded).

**Table 3 | Cox regression models for overall survival for 2.5 year landmark (n = 138)**

| Covariate | Detail | p value | HR | 95% CI |
|---|---|---|---|---|
| **Refined model (p = 0.0001, Chi-squared = 23.19): OS for 2.5 y landmark (n = 138) with study Arm** | | | | |
| Study Arm | A (12MP+Tet) vs. D (12MP + 6MHP+Cy) | **0.019*** | 2.915 | 1.196 to 7.092 |
| Age | ≤40 vs >40 years | **0.025*** | 0.103 | 0.014 to 0.751 |
| ECOG PS score | 1 vs 0 | **0.014*** | 2.855 | 1.233 to 6.614 |
| Advanced disease status | Recurrence vs initial diagnosis | *0.074*** | 0.561 | 1.233 to 1.057 |
| **Refined model (p = 0.0002, Chi-squared = 22.18): OS for 2.5 y landmark (n = 138) with vaccine regimen and Cy arms** | | | | |
| Vaccine regimen[a] | 12MP + tet vs. 12MP + 6MHP | **0.017*** | 2.208 | 1.153 to 4.228 |
| Age | ≤40 vs >40 years | **0.027*** | 0.105 | 0.014 to 0.771 |
| ECOG PS score | 1 vs 0 | **0.044*** | 2.336 | 1.025 to 5.323 |
| Advanced disease status | Recurrence vs initial diagnosis | *0.062*** | 0.544 | 0.287 to 1.031 |

*HR* hazard ratio, *CI* confidence interval, *ECOG PS* Eastern Cooperative Oncology Group Performance Status, *12MP* 12 class I MHC-restricted melanoma peptides, *tet* tetanus toxoid helper peptide,
*6MHP* mixture of six melanoma-specific helper peptides, *LDH* lactate dehydrogenase.
*Significant, *p* < 0.05 (bolded); **0.1 > *p* > 0.05 (italics).
[a]Vaccine regimen 12MP + tet includes study arms A + B; vaccine regimen 12MP + 6MHP includes study arms C + D.

rank test comparing OS estimates favored the 12MP + 6MHP vaccine regimen (*p* = 0.08), but early overlap of the OS curves represents a challenge for statistical modeling, as has been observed in other immunotherapy trials[17–21]. Nonetheless, Cox regression analyses for the ITT population included four significant covariates: study arm (A vs D, *p* = 0.0180), AJCC stage (*p* = 0.048), age (*p* = 0.003), and sex (*p* = 0.032, Table 2). When study arm was not included in the multivariable model, but instead vaccine regimen (6MHP yes/no) and Cy arms (Cy yes/no) were included, there were trends (*p* ~ 0.08) for better OS with 6MHP vaccines and with Cy (Supplementary Table 4). Together, these data support sex and age as significant covariates for OS and RFS. For OS, there are trends for the vaccine regimen (6MHP arms) and the Cy arms to be associated with favorable outcome, which is consistent with the univariate *p* values for the impact of vaccine regimen. However, the inclusion of study arm allows comparison of outcomes for arm D vs arm A, and reveals a significant association with better outcome for arm D both for OS and for RFS. This may reflect the combined benefit of adding 6MHP and Cy in arm D, which is not captured by either vaccine regimen or by Cy arms.

Most time-to-event analyses use the proportional hazards assumption, meaning that the hazard rate is assumed to remain constant throughout the follow-up period. The observed delay in separation of the survival curves suggests non-proportionality, which may complicate testing significance of the difference between OS curves with a traditional log-rank test. An approach suggested for testing differences in such cases in immuno-oncology is to analyze the hazard ratio after separation of the curves[19,20,22]. Using this approach, with a landmark at 2.5 years, there was a significant prolongation of survival for participants on the 12MP + 6MHP vaccine regimen (HR 0.52; *p* = 0.04, Fig. 2C), which was supported in multivariable Cox regression analyses, where vaccine regimen was the most significant predictor of OS (Table 3). Taken together, these data suggest a meaningful enhancement of OS by inclusion of the melanoma-cognate 6MHP vaccine peptides, and likely by addition of Cy pretreatment. These findings provide justification for considering a larger randomized trial, powered to detect improved survival. We estimate that 309 patients per arm would be required to power a two-arm randomized trial for HR 0.65, alpha 0.05, power of 0.8.

The RFS data from this study also favor the 12MP + 6MHP vaccine regimen, though with less separation than the OS curves. This may be explained in part by differences in resectable regional recurrences as opposed to unresectable distant metastases, for which future analyses are warranted. Perhaps the most striking feature of the RFS curves is their flattening after 4 years, with about 50% recurrence-free long-term after 12MP + 6MHP vaccines and 40% recurrence-free after 12MP+tet vaccines (Fig. 2D). These suggest durable disease control with vaccine-monotherapy for both vaccine regimens, with or without Cy, for these high-risk patients after surgery. OS estimates continue to decline through about 14 years of followup (Fig. 2A); however, OS rates at 15 years still exceed the RFS rates at that time. For patients with resectable recurrences, some may experience long-term disease-free survival. Also, subsequent therapy with checkpoint blockade agents or BRAF/MEK inhibition for recurrences during the follow-up period can lead to delayed mortality over a period of years so that the OS curves gradually approach the long-term RFS curves.

The 12MP and 6MHP vaccines target shared melanoma antigens, including both melanocytic differentiation proteins (MDPs: gp100, tyrosinase, and MART-1/MelanA) and cancer-testis antigens (CTAs: MAGE antigens and NY-ESO-1). Prior vaccine trials using just class I MHC-restricted peptides from these proteins have been generally disappointing, as have whole protein vaccines[23–25]. This has led to common conclusions that MDPs and CTAs may not be effective tumor-rejection antigens and that vaccines targeting them may not be effective. The data presented in this manuscript provide support for the relevance and effectiveness of targeting these shared antigens, when stimulating CD4$^+$ and CD8$^+$ T cells with the same vaccine. The relevance of gp100 as a tumor-rejection antigen is also supported by survival benefit and FDA approval of treating metastatic uveal melanoma with tebentafusp, a bispecific T cell engager built on an enhanced T cell receptor binding gp100$_{280-288}$[26–28]. This peptide target, originally described by our group[29], is included in 12MP. Another gp100 peptide in 12MP is gp100$_{209-217}$(210 M), which improved objective response rates and progression free survival outcomes when added to systemic interleukin-2 therapy in advanced melanoma[30], despite failing to add benefit to ipilimumab in a later trial[31]. Also, adoptive T cell therapies targeting MelanA/MART-1, gp100, and NY-ESO-1 have induced objective clinical responses in melanoma and other cancers, supporting these as cancer-rejection antigens[32–34]. More recently, an mRNA vaccine strategy using 4 non-mutated antigens, including tyrosinase, MAGE-A3, and NY-ESO-1, has induced durable objective tumor responses[35–37]. Thus, there is a body of literature supporting the value of targeting shared non-mutated melanoma antigens. The present data from a randomized clinical trial support enhanced overall survival with a melanoma vaccine targeting shared antigens in the adjuvant setting, and they support a therapeutic advantage of adding melanoma-specific helper peptides.

In our initial report of this trial, immune responses to peptides in the vaccines were reported, based on ELIspot assays for interferon-gamma (IFNγ)[10]. Briefly, circulating helper T cell responses were detected ex vivo in response to the tetanus helper peptide in 91% of patients for Arms A and B, and in response to 6MHP in 52% of patients for Arms C and D[10]. CD8$^+$ T cell responses were detected more often for Arms A and B than Arms C and D[10]. The encouraging survival outcomes of patients vaccinated with 12MP + 6MHP vaccines is surprising given these lower CD8$^+$ T cell responses. An understanding of the immunologic basis of clinical benefit of vaccines including cognate T cell help likely will require understanding more about T cell homing and effects on myeloid cells in the tumor microenvironment. Addition of cognate T cell help in a murine study has enhanced T cell infiltration by CD8$^+$ T cells[2]. CD4$^+$ T cells also can enhance tumor control by modulating macrophage function[38]. Thus, future goals are to gain a more detailed understanding of multifunctional CD4$^+$ and CD8$^+$ T cell responses, induction of stem-like memory and T cell homing receptor expression

on vaccine induced T cells, durable T cell responses in long-term survivors, and evaluation of T cells infiltrating recurrent tumors after vaccination.

Cy pretreatment was tested in this study based on prior work showing reduction of regulatory T cells and improved T cell responses and/or tumor control when combined with cancer vaccines in murine models and in non-randomized clinical trials[11–16]. Cy may enhance immunogenicity through inhibition of nitric oxide synthase[15]. The current trial used a single dose of Cy 5 days pre-vaccine, which was well-tolerated[10]. In the original report, it did not alter T cell response rates to the peptide vaccines[10], and in this long-term followup of the Mel44 trial, univariate analyses did not detect a significant impact on OS, and only a weak trend favoring prolonged RFS. However, the Cox regression analyses identify Arm A patients as those with the lowest OS and Arm D as most favorable, suggesting a potential clinical benefit of this Cy pretreatment when combined with 12MP + 6MHP. This is particularly evident for OS in the 2.5-year landmark analysis, where the HR between Arms D and A is 0.40 (95% CI 0.16–0.99, Supplementary Fig. 2C) and for RFS for male patients where the HR is 0.42, favoring the Cy group (95% CI 0.2–0.89, Supplementary Fig. 5B). Additional studies of Cy pretreatment may be warranted in combination with the 12MP + 6MHP vaccine.

Perhaps the most unexpected finding was a sex-dependence of clinical benefit by adding 6MHP. OS for males receiving 12MP + 6 MHP vaccines was more favorable than OS for males receiving 12MP +tet and for all female participants. In the 2.5 landmark analysis, sex was not a significant covariate for OS, but even in that dataset, males who received the 6MHP vaccines had more favorable survival, and the HR for benefit of the 6MHP vaccines was more favorable for males than females. When controlling for covariates, in a Cox regression analysis for OS among male participants, vaccination with 6MHP was significantly associated with OS ($p = 0.022$). Similarly, Arm D was associated with significantly better OS than Arm A when the 4 arms were included in the model. Also, when evaluating just the participants who received 12MP + 6MHP (Arms C + D), a regression model for OS identified sex as the only significant co-variate. Multi-variable regression modeling did not identify significant interactions between sex and vaccine regimen, but trended to significance for RFS for the ITT population (0.081); however, the sample size required to demonstrate a significant interaction is about 4x that required for the main effects[39]. Overall, our findings identify sex as a key covariate in OS and RFS and specifically impacting the outcomes after vaccination with 12MP + 6MHP. This is surprising because women with melanoma are less likely to develop metastatic disease[40] and more likely to experience longer OS[40–42]. Women often have stronger immune responses to vaccines[43] and are more prone to autoimmune diseases[44], likely due to stronger antibody and Th2 T cell responses. However, males may have more Th1-dominant responses[45]. Also, estradiol has been reported to drive macrophages to more immunosuppressive states through interaction with estrogen receptor alpha, turning them to M2 macrophages, and blocking estrogen can enhance tumor control by PD-1 blockade[46]. Interestingly, multiple trials of PD-1 blockade have revealed higher objective response rates in men than women[47–49]. Some work has identified genetic signatures with sex-associated disparate effects on clinical response to checkpoint blockade therapy[50]. Our results support further investigation into the differences in clinical outcomes by sex, including evaluating differences in immune responses and the tumor microenvironment by sex, and consideration of using different immune therapy combinations based on patient sex.

Recent findings from a phase II trial of a neoantigen mRNA vaccine (NCT03897881) found preliminary evidence for prolonged RFS and distant metastasis-free survival with that vaccine (mRNA-4157) for patients with stage III-IV melanoma in the adjuvant setting when combined with pembrolizumab, supporting a future definitive phase

III trial testing that approach[51,52]. Vaccines targeting non-mutated tumor antigens may further enhance cancer vaccine activity. They have an advantage of being off-the-shelf vaccines that can be administered without the delay inherent to preparing personalized neoantigen vaccines. As data have emerged showing benefit of neoadjuvant systemic therapy prior to surgery, with checkpoint blockade or BRAF/MEK inhibition[53,54], it is appealing to consider vaccine combinations with those agents in the neoadjuvant setting. Murine data, and our own data in humans, suggest that combining vaccines with PD-1 blockade may be most beneficial if both are begun concurrently rather than adding vaccines after prior PD-1 blockade[55,56]. Thus, vaccines targeting shared non-mutated antigens may be the best options for combination with neoadjuvant PD-1 blockade therapy. There may also be benefit of combining shared non-mutated antigens with mutated neoantigen vaccines, either concurrently or sequentially, whether using peptides or mRNA formulations.

There are some limitations to acknowledge in this manuscript. The study design was not powered for OS or RFS. However, clinical outcome was an exploratory endpoint, and the present report does support prolongation of OS by addition of cognate helper peptides instead of non-specific help. Also, patients were enrolled and treated from 2005 to 2010, when it was not routine to test for BRAF mutation status, PD-L1 expression, or tumor mutation burden. Thus, these data are not available; however, the study randomization led to well-matched groups; so, we expect balances in these features. BRAF mutations are inversely associated with patient age[57], and study arms were well matched for age. Multivariable analyses did support the univariate analyses, and they provide motivation now to uncover mechanisms for the improved outcomes for patients vaccinated with 6MHP in the absence of better CD8+ T cell responses, and for the improved impact of 6MHP in males compared to females. We anticipate that these will be elucidated by more comprehensive analyses of immune responses and the impact of sex on the melanoma tumor microenvironment.

In summary, long-term follow-up of patients on the Mel44 vaccine trial reveals a durable prolongation of OS by adding cognate helper peptides to a vaccine otherwise designed to stimulate CD8+ T cells. These data also provide support for clinical benefit of targeting non-mutated shared antigens. Benefit is most evident in earlier stage patients and appears confined to males. Benefit of Cy pretreatment is not observed in the primary comparison, but evaluation across all 4 arms, especially in males, leaves open the possibility that Cy pretreatment may be helpful when combined with cognate T cell help. The results support design of future trials powered to evaluate long-term clinical outcomes with 12MP + 6MHP vaccination plus PD-1 blockade vs PD-1 blockade alone, in the adjuvant setting, stratified by sex, particularly in a trial that would use a biomarker to identify high-risk patients.

## Methods

### Patient selection

This research study complies with all relevant ethical regulations: the clinical trial MEL44 was performed with approval of the institutional review boards (IRB) at the 3 participating institutions (University of Virginia, MD Anderson Cancer Center at the University of Texas – Houston, and Fox Chase Cancer Center), with the University of Virginia at the lead institution (IRB-HSR #11491). It was also performed with FDA approval (IND #12191) and is registered with ClinicalTrials.gov (NCT00118274). Participants gave informed consent prior to participating in this trial. The study design and conduct complied with all relevant regulations regarding the use of human study participants and was conducted in accordance with the criteria set by the Declaration of Helsinki, as represented by the IRB-HSR approval at the University of Virginia and the other two participating institutions, and monitoring by the University of Virginia School of Medicine Clinical

Trials Office and the University of Virginia Cancer Center Data Safety Monitoring Committee.

This study included participants followed for long-term outcomes enrolled in the Mel44 (NCT00118274) clinical trial, a multicenter, randomized phase II trial designed to evaluate immunologic and clinical outcomes of patients vaccinated with 12 Class I MHC-restricted peptides (12MP) to stimulate melanoma-reactive CD8+ T cells plus one of two "helper" peptide preparations to stimulate CD4+ T cells, either non-specific help (tet) or melanoma-specific help (6MHP) (Supplementary Table 1), in an water-in-oil emulsion with an incomplete Freund's adjuvant (Montanide ISA-51, Seppic, Inc, Courbevoie, France). The full protocol is provided as Supplementary Note in the Supplementary Information, and the registration on ClinicalTrials.gov (posted 11 July 2005) is available at https://www.clinicaltrials.gov/study/NCT00118274. Patients with resected stage IIB-IV melanoma by American Joint Committee on Cancer (AJCC; 6th edition)[58] arising from cutaneous, mucosal, or unknown primary sites were eligible for inclusion. Eligibility included expression of at least one of the following Class I MHC molecules HLA-A1, -A2, or -A3; and at least one of the following Class II MHC molecules HLA-DR1, -DR4, -DR11, -DR13, or -DR15, as restricting MHC for the peptides in the vaccines. Complete inclusion and exclusion criteria have been previously defined[10]. The present study included all eligible patients enrolled in Mel44. The first and last patients were enrolled May 2005 and February 2008, respectively.

### Clinical trial design

Trial design and the clinical trial protocol have been previously described[10] and are represented in Fig. 1A. The study was originally designed and powered to assess safety and immune responses[10]. Target enrollment was 40 participants per arm (total 160). At final analysis, 167 eligible participants were enrolled and treated[10]. For the present analysis of long-term clinical outcome, the sample size was the population of 167 eligible participants. Vaccines were administered on weeks 0, 1, 2, 4, 5, 6, 12, 26, 39, and 52. Participants were randomized 1:1:1:1 to one of four study arms to receive one of two peptide vaccine formulations, with or without single intravenous dose of cyclophosphamide (Cy, 300 mg/m²) 5 days prior to the first vaccine. Participant enrollment was stratified by Class I HLA type and participating institution. A 2×2 design was used, to enable comparison of pairs of arms: A + B vs C + D for the vaccine regimen comparison (Fig. 1B), and A + C vs B + D for the Cy comparison. For this study, long-term outcomes were analyzed by arm and by grouping arms based on helper peptide regimen and cyclophosphamide regimen (Fig. 1).

### Clinical data and clinical outcomes

Baseline clinical data were retrieved from the Cancer Center Clinical Trials Office database and from the OnCore database and associated clinical records. For the present analysis, AJCC staging was converted to AJCC version 8 (v8). All participants were followed for survival and disease recurrence by clinicians and clinical research staff at their treating institutions. Data on date of last known follow-up, date of disease recurrence, date of last known disease status, and date of death from any cause were obtained from study case report forms. Endpoints for this analysis were OS and RFS. OS was measured from study entry to last known follow-up or date of death. RFS was measured from study entry to date of disease recurrence, including new primary melanomas, or date of last known disease status[59].

### Statistical analyses

Summary statistics for participant characteristics were reported using medians and ranges for continuous variables and using counts and percentages for categorical variables. Participant characteristics were

compared between study arms using the Mann–Whitney test for continuous variables and using Chi-square test or Fisher's exact test as appropriate for categorical variables. OS and RFS were estimated using the Kaplan–Meier method and compared among vaccine regimens using the log-rank test and Cox regression analysis. The median survival time and estimated survival probability were reported. Hazard ratios (HR) and 95% confidence intervals from Cox regression analyses were used to quantify the difference in OS or RFS between regimens. When Kaplan–Meier curves overlapped early, HRs were also assessed after that early period of overlap, as has been recommended for immunotherapy trials[19,20]. Milestone outcomes[60] are assessed at 5 year intervals. Exploratory subgroup analyses were performed for the impact of patient stage (II-III vs IV), age, and biologic patient sex may contribute to outcomes. Ages were defined by rounding down to the nearest year. Two-sided $p$-values were reported and a $p$-value less than 0.05 was considered statistically significant. Fisher's exact tests larger than 2 × 2 were calculated using R version 4.2.2 and RStudio v2022.12.0 (Build 353). All other statistical analyses were performed using MedCalc® Statistical Software version 22.016 (MedCalc Software Ltd, Ostend, Belgium; https://www.medcalc.org; 2023). ChatGPT provided R code for Fisher's exact tests, for information on suggested sample sizes for a future clinical trial to test the impact of the 12MP + 6MHP vaccine definitively, and for using the pwr package in R.

### Reporting summary
Further information on research design is available in the Nature Portfolio Reporting Summary linked to this article.

## Data availability
Source data are provided with this paper and include individual de-identified participant data (some of the variables are provided in aggregate since consent to publish clinical information potentially identifying individuals was not obtained). Additional individual de-identified participant data can be shared upon request to Dr. Slingluff. The clinical trial protocol is also provided as Supplementary Note in the Supplementary Information. The remaining data are available within the Article, Supplementary Information or Source Data file. Source data are provided with this paper.

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

## Acknowledgements

We are grateful to all the clinical research coordinators at all participating sites who collected and recorded survival followup data for this study over 15 years. The trial was supported by NCI R01 CA118386 (C.L.S.); P30 CA044579 (Molecular and Immunologic Translational Sciences Core, Biorepository and Tissue Research Facility, Office of Clinical Research, and Biostatistics Shared Resource); gifts from Alice and Bill Goodwin and the Commonwealth Foundation for Cancer Research. C.L.S. has served as sponsor-investigator, holding the IND on behalf of the University of Virginia. There was no other external sponsor.

## Author contributions

Long-term follow-up data were collected under the supervision of C.L.S., K.A.C.-B., M.V.M., N.B.H. and M.I.R. E.K.N. prepared the initial manuscript draft and edited extensively. All authors have reviewed and contributed to the manuscript writing. E.K.N., L.T.D., and C.L.S. reviewed and aggregated long-term follow-up data, and clinical data, and prepared tables and figures. Statistical review and reporting were provided by C.L.S. and H.Z.

## Competing interests

C.L.S. has the following disclosures: Research support to the University of Virginia from Celldex (funding, drug), Glaxo-Smith Kline (funding), Merck (funding, drug), 3 M (drug), Theraclion (device staff support); Funding to the University of Virginia from Polynoma for PI role on the MAVIS Clinical Trial; Funding to the University of Virginia for roles on

Scientific Advisory Boards for Immatics and CureVac. Also, C.L.S. has received licensing fee payments through the UVA Licensing and Ventures Group (UVA LVG) for patents for peptides used in these cancer vaccines. Patents relevant to peptides in this vaccine include United States Patent # 6,660,276; 6,558,671; and 7,019,112; however, the terms of those patents have expired. He holds other patents for peptides in melanoma vaccines (US Patent number US 9,345,755 B2, plus one submitted, all managed by UVA LVG. N.B.H. does not have any conflicts relative to melanoma research, but has attended advisory board meetings for Eisai, BMS, Merck, Exelixis in the past 2 years. M.I.R. has relationships with Merck (US Global Advisory Board Member (<$10,000) and Consultant (Travel Expenses) and Amgen (Consultant (travel expenses) and Support to the institution for clinical research support). There are no other disclosures or competing interests for the other authors.
