## [Peer Review File · Nature Communications]

Multipeptide vaccines for melanoma in the adjuvant setting:
long-term survival outcomes and exploratory analysis of a
randomized phase II trialEditorial Note: This manuscript has been previously reviewed at another journal that is not operating a transparent peer review scheme. This document only contains reviewer comments and rebuttal letters for versions considered at *Nature Communications*. Mentions of the other journal have been redacted.

REVIEWER COMMENTS

Reviewer #2 (Remarks to the Author):

The authors have addressed my previous comments for [journal name redacted]. I have no further comments or suggestions.

Reviewer #4 (Remarks to the Author):

This manuscript reported the long term follow up results of the Mel44 trial, in which clinical outcomes OS and RFS were compared in advanced melanoma patients who received vaccine 12MP +/- 6MHP, and +/- cy. This is certainly a valuable report that warrant its impact in the clinical community for melanoma and beyond.

Although only borderline significance was achieved in OS (insufficient power as discussed by the authors), the overall survival benefit from OS and RFS, especially after taking into account the delayed benefit from immunotherapy, is convincing on the superiority of including the 6MHP helper together with the CD8 targeting vaccine. The multivariable analysis is important to establish this conclusion. Overall, I found this manuscript well written, with analysis and discussion appropriate for the conclusions. The comments from previous reviewers have been properly addressed within the scope of this manuscript.

There remain several conclusions that could be further addressed by additional data analysis.

1. One major conclusion in this study is the superior survival benefit from 6MHP vaccine in male only, which was clearly illustrated in K-M curves. A multivariable Cox regression analysis is recommended to include an interaction between gender and 6MHP yes/no, so

the magnitude of survival benefit in gender and 6MHP+ arms can be quantified. At the moment, multivariable analysis is carried out on patient sub-cohorts. This analysis should also be repeated for OS after 2.5 years landmark and RFS. At the moment, the OS benefit in male disappeared after 2.5 years, which is worrying and has not been discussed.

2. It is unusual to observe survival benefit in stage II, III patients but not in stage IV patients. It is worth checking the clinical features associated with stage IV and address in a multivariable model whether the unusual survival benefit from stage can be explained by the concept of co-linearity.

3. The manuscript highlighted the comparison of arm A and D, hypothesising accordingly that Cy should be administered with 6MHP. However, there has been criticism from previous reviewers challenging the reliability of this hypothesis. The 2*2 design assumes independent / additive treatment benefit from 6MHP and Cy. An exploratory multivariable analysis can therefore be carried out to strengthen the hypothesis. I recommend that the treatment arm be represented using two binary variables 6MHP yes/no and Cy yes/no, so the survival benefit from each treatment can be quantified.

Minor comments:

1. Label the treatment arms in a consistent way across all figures, e.g., arm a,b,c,d vs. 12MP + tet.

2. Label the figure with p-values even if it is not significant and therefore not quoted in the main text.

3. Change “multivariate analysis” into “multivariable analysis”. The former refers to analysing multiple dependent variables in statistics.

Point-by-point reply to reviewer comments:

“Survival benefit of adding tumor-cognate helper peptides to a melanoma vaccine: dependent on patient sex.”

We thank the reviewers for the favorable review from and positive comments. We appreciate the suggestions for additional analyses and revisions from reviewer #4. They are addressed point-by-point below, and the manuscript has been revised accordingly.

Reviewer comment #1. One major conclusion in this study is the superior survival benefit from 6MHP vaccine in male only, which was clearly illustrated in K-M curves. A multivariable Cox regression analysis is recommended to include an interaction between gender and 6MHP yes/no, so the magnitude of survival benefit in gender and 6MHP+ arms can be quantified. At the moment, multivariable analysis is carried out on patient sub-cohorts. This analysis should also be repeated for OS after 2.5 years landmark and RFS. At the moment, the OS benefit in male disappeared after 2.5 years, which is worrying and has not been discussed.

Response (in three parts: I, II, III):

I. Additional multivariable regression analyses for OS for the full dataset. We have run a multivariable Cox regression analysis for OS for the full dataset (n = 167) including the sex and 6MHP yes/no, as well as multiple other clinical variables. For this analysis, we also incorporated the recommendation from comment #3, to include separate variables for 6MHP (yes/no) and Cyclophosphamide (yes/no). The initial model with all 167 patients had an overall p value of <0.0001, provided in the table below, where significant associations were found with age (<=40 vs > 40 years) and sex, with p values approaching significance for 6MHP (yes/no) and Cy (yes/no).

Initial model (p < 0.0001, Chi-square 36.59): OS all eligible patients (n = 167) Including vaccine regimen (6MHP yes/no) and Cyclophosphamide (yes/no), not study arms				
Covariate	Detail	p value	HR	95% CI
Age	≤ 40 vs > 40 years	0.0026*	0.0464	0.0063 to 0.3413
Sex	Female vs male	0.0243*	1.8165	1.081 to 3.054
Vaccine Regimen: 6MHP (yes/no)	12MP+tet vs. 12MP+6MHP	0.0723	1.5797	0.9594 to 2.601
Cyclophosphamide Arms: Cy (yes/no)	No Cy vs. +Cy	0.0773	1.5665	0.9521 to 2.5774
Advanced disease status	Recurrence vs initial diagnosis	0.2526	0.7399	0.4416 to 1.2397
LDH level	High vs normal or not done	0.2312	2.1004	0.6233 to 7.0779
AJCC stage, v8	Stage IV vs II-III	0.1942	0.6291	0.3125 to 1.2665
ECOG PS score	1 vs 0	0.3825	1.3673	0.6776 to 2.7592

These two significant covariates (age and sex) were the same as reported in **Table 2** when using study arm as a covariate, with better OS for younger and male patients. However, the vaccine regimen (6MHP yes/no) and Cy arms (yes/no) approach significance, whereas that study arm covariate (arm D vs arm A) was significant in our model reported in **Table 2**.

When we refined the new model above by excluding covariates with high p values, the refined model overall still has p < 0.0001, with the following details:

Refined model (p < 0.0001, Chi-squared 33.324): OS all eligible patients (n = 167) Including vaccine regimen (6MHP yes/no) and Cyclophosphamide (yes/no), not study arms				
Covariate	Detail	p value	HR	95% CI
Age	≤ 40 vs > 40 years	0.0029*	0.0492	0.0068 to 0.3579
Sex	Female vs male	0.0337*	1.7361	1.0436 to 2.8881
AJCC stage, v8	Stage IV vs II-III	0.0474*	0.5187	0.2710 to 0.9925
Vaccine Regimen: 6MHP (yes/no)	12MP+tet vs. 12MP+6MHP	0.0820	1.5494	0.9459 to 2.5381
Cyclophosphamide Arms: Cy (yes/no)	No Cy vs. +Cy	0.0826	1.5469	0.9452 to 2.5316

We ran the model again with all the same starting covariates and also adding in the study arm, followed by stepwise optimization (keep covariate if $p < 0.1$), the following model was generated:

Optimized model ($p < 0.0001$, Chi-squared 32.172): OS all eligible patients ($n = 167$) Including vaccine regimen (6MHP yes/no) and Cyclophosphamide (yes/no), and study arms				
Covariate	Detail	p value	HR	95% CI
Age	≤ 40 vs > 40 years	0.0027*	0.0483	0.0066 to 0.3508
Sex	Female vs male	0.0182*	1.8169	1.1068 to 2.9828
Study Arm	A (12MP+Tet) vs D (12MP+6MHP+Cy)	0.0252*	1.8474	1.0792 to 3.1623
AJCC stage, v8	Stage IV vs II-III	0.0537	0.5286	0.2766 to 1.0103

Inclusion of interaction variable for Sex and Vaccine regimen. The reviewer recommended assessment of interaction between Sex and vaccine regimen. The refined model (bottom page 1) includes Sex and Vaccine regimen; so, we repeated the Cox regression analysis with these covariates after adding an interaction term for Sex and Vaccine regimen. The result was:

Refined model ($p < 0.0001$, Chi-squared 35.222): OS all eligible patients ($n = 167$) Including vaccine regimen (6MHP yes/no) and Cyclophosphamide (yes/no), not study arms Plus addition of an interaction term				
Covariate	Detail	p value	HR	95% CI
Age	≤ 40 vs > 40 years	0.0025*	0.0467	0.0064 to 0.3397
Sex	Female vs male	0.4593	1.2870	0.6596 to 2.5112
AJCC stage, v8	Stage IV vs II-III	0.0257*	0.4690	0.2411 to 0.9122
Vaccine Regimen: 6MHP (yes/no)	12MP+tet vs. 12MP+6MHP	0.9598	0.9797	0.4406 to 2.1783
Sex*Vaccine Regimen	Interaction between Sex and Vaccine regimen	0.1651	0.4827	0.1726 to 1.3499
Cyclophosphamide Arms: Cy (yes/no)	No Cy vs. +Cy	0.1116	1.4928	0.9112 to 2.4456

The interaction term has a p value of 0.1651 in this model, indicating a non-significant interaction effect. Thus, we would remove the interaction term from the refined model. We have similarly added the interaction term into models for RFS for the ITT dataset and for OS for the 2.5 year landmark data, and the p values for the interaction term were 0.081 and 0.207, respectively.

We have modified the Results section, first to update some details in the original model in the following paragraph:

For OS, these variables defined a significant model with 3 significant covariates: study arm, sex, and age. When covariates with p values >0.1 were removed, the final model included the same 3 significant covariates: age (≤40 vs >40 years, HR 0.049, $p = 0.003$), study arm (A vs D, HR 2.362, $p = 0.018$) and sex (female vs male, HR 1.755, $p = 0.032$; **Table 2**).

Following that, we added the following paragraph summarizing the information above, and we reference a new Supplemental Table 4, with the details of those models:

In that analysis, vaccine regimen and Cy arms were not significant when study arm was included as a covariate. Thus, we also performed multivariable analysis with the vaccine

regimen, Cy arms, and the same 6 clinical covariates, but without study arm. A refined model was similar with significantly better survival with younger age ($p = 0.003$), male sex ($p = 0.034$), and AJCC stage II-III ($p = 0.047$), and with trends to better survival with regimens containing 6MHP ($p = 0.082$) and Cy ($p = 0.083$, **Supplemental Table 4**).

Text is added to the end of paragraph 2 of the Discussion as well:

When study arm was not included in the multivariable model, but instead vaccine regimen (6MHP yes/no) and Cy arms (Cy yes/no) were included, there were trends ($p \sim 0.08$) for better OS with 6MHP vaccines and with Cy (**Supplemental Table 4**). Together, these data support sex and age as significant covariates for OS and RFS. For OS, there are trends for the vaccine regimen (6MHP arms) and the Cy arms to be associated with favorable outcome, which is consistent with the univariate p values for the impact of vaccine regimen. However, the inclusion of study arm allows comparison of outcomes for arm D vs arm A, and reveals a significant association with better outcome for arm D both for OS and for RFS. This may reflect the combined benefit of adding 6MHP and Cy in arm D, which is not captured by either vaccine regimen (Arms A+B vs C+D) or by Cy arms (A+C vs B+D).

To address the tests of interactions: text is added to the Results (end of 3rd to last paragraph):

We did explore if there is an interaction between sex and vaccine regimen, by adding an interaction term (Sex*Vaccine regimen), to the models for OS and RFS for the ITT population, and for OS for the 2.5 year landmark population: those p values for the interaction term were 0.165, 0.081 and 0.207, respectively.

and we added to the Discussion (3rd from the last paragraph)

Multivariable regression modeling did not identify significant interactions between sex and vaccine regimen, but trended to significance for RFS for the ITT population (0.081); however, the sample size required to demonstrate a significant interaction is about 4x that required for the main effects {Leon, 2009 #16384}.

II. Additional multivariable regression analyses for RFS for the full dataset. We have also performed similar multivariable analyses for RFS, overall ($n = 167$). The initial model is shown below, with only Age significant, but sex approaching significance.

Initial model ($p = 0.0589$, Chi-squared 15.015): RFS, all eligible patients ($n = 167$)				
Covariate	Detail	p value	HR	95% CI
Age	≤ 40 vs > 40 years	0.0291*	0.4500	0.2196 to 0.9221
Sex	Female vs male	0.0630	1.5168	0.9777 to 2.3530
Vaccine Regimen: 6MHP (yes/no)	12MP+tet vs. 12MP+6MHP	0.3080	1.2452	0.8168 to 1.8983
Cyclophosphamide Arms: Cy (yes/no)	No Cy vs. +Cy	0.1107	1.4187	0.9231 to 2.1804
Advanced disease status	Recurrence vs initial diagnosis	0.1107	1.4439	0.9194 to 2.2678
LDH level	High vs normal or not done	0.7204	1.2443	0.3760 to 4.1175
AJCC stage, v8	Stage IV vs II-III	0.6796	0.8947	0.5276 to 1.5171
ECOG PS score	1 vs 0	0.8449	1.0742	0.5243 to 2.2007

Then, we ran the model with stepwise selection of covariates, leading to a significant model with more limited covariates, but with age and sex only approaching significance.

Initial model (p = 0.0106, Chi-squared = 11.226); RFS all eligible patients				
Covariate	Detail	p value	HR	95% CI
Age	≤ 40 vs > 40 years	0.0501	0.4967	0.2466 to 1.0005
Sex	Female vs male	0.0749	1.4776	0.9616 to 2.2706
Advanced disease status	Recurrence vs initial diagnosis	0.0918	1.4427	0.9422 to 2.2090

Alternatively, when we included all those 8 covariates plus also the covariate for study arm, and select stepwise, a very significant model overall is generated, with significance for age and sex, and p = 0.0556 for arm A vs. D.

Revised model with Study Arm (p = 0.0055, Chi-squared = 14.656): RFS all eligible patients, n = 167				
Covariate	Detail	p value	HR	95% CI
Age	≤ 40 vs > 40 years	0.0266*	0.4459	0.2183 to 0.9106
Sex	Female vs male	0.0485*	1.5461	1.0029 to 2.3834
Study Arm	A (12MP+Tet) vs D (12MP+6MHP+Cy)	0.0556	1.5888	0.9890 to 2.5521
Advanced disease status	Recurrence vs initial diagnosis	0.1367	1.3841	0.9021 to 2.1236

Since the 4th variable in that model was not a significant covariate, we created the model without it, which also is highly significant and includes Age, Sex, and Study Arm as significant covariates:

Final model with Study Arm (p = 0.0061, Chi-squared = 12.417): RFS all patients, n = 167				
Covariate	Detail	p value	HR	95% CI
Age	≤ 40 vs > 40 years	0.0136*	0.4100	0.2019 to 0.8323
Sex	Female vs male	0.0335*	1.5960	1.0372 to 2.4557
Study Arm	A (12MP+Tet) vs D (12MP+6MHP+Cy)	0.0363*	1.6536	1.0325 to 2.6481

We have added this final model in **Supplemental Table 5**, which is modified (prior Supplemental Table 4). We limited the values in the table to 3 decimal places, to be consistent with the rest of the manuscript. We also added a new **Supplemental Table 6** that includes the models above that included vaccine regimen and Cy arms, but not study arm. We have modified the relevant paragraph in the Results: the first sentence, below, has been modified to correct the p values to match the final model. We added a second sentence that explains the findings in **Supplemental Table 6**.

For RFS, the same starting covariates, we removed those with adjusted p values > 0.1, leaving age, study arm, and sex. This model for RFS (p = 0.006) was driven by age (p = 0.014), sex (p = 0.034) and study arm (p = 0.036; **Supplemental Table 5**). We also ran multivariable analysis for RFS excluding arm, and no covariates were significant, but age, sex, and advanced disease status approached significance (**Supplemental Table 6**).

We added text to the first paragraph of the Discussion as well:

When study arm was not included in the multivariable model, but instead vaccine regimen (6MHP yes/no) and Cy arms (Cy yes/no) were included, there were trends (p ~0.08) for better OS with 6MHP vaccines and with Cy (**Supplemental Table 4**).

III. Multivariable regression analyses for 2.5 year landmark data. We have performed multivariable regression analysis for the 2.5 year landmark OS data. Considering the 6 clinical covariates (AJCC stage, Age, Sex, ECOG PS, advanced disease status, and LDH level) plus the vaccine regimen (6MHP yes/no) and the Cy arms (yes/no), for these 138 patients, a significant model was generated, with significance for Age and Vaccine regimen.

Initial model (p = 0.0007, Chi-squared = 27.138): OS for 2.5y landmark (n = 138)				
Covariate	Detail	p value	HR	95% CI
Vaccine Regimen: 6MHP (yes/no)	12MP+tet vs. 12MP+6MHP	0.0324*	2.0659	1.0629 to 4.0153
Cyclophosphamide Arms: Cy (yes/no)	No Cy vs. +Cy	0.1085	0.5870	0.3062 to 1.1251
AJCC stage, v8	Stage IV vs II-III	0.9424	0.9688	0.4101 to 2.2889
Age	≤ 40 vs > 40 years	0.0093*	0.0652	0.0083 to 0.5108
Sex	Female vs male	0.1837	1.5993	0.8003 to 3.1958
ECOG PS score	1 vs 0	0.1226	1.9788	0.8320 to 4.7060
Advanced disease status	Recurrence vs initial diagnosis	0.0567	0.5139	0.2591 to 1.0193
LDH level	High vs normal or not done	0.1080	3.5378	0.7579 to 16.5152

After stepwise selection, a more significant model, with 4 variables resulted, including vaccine regimen:

Refined model (p = 0.0002, Chi-squared = 22.182): OS for 2.5y landmark (n = 138)				
Covariate	Detail	p value	HR	95% CI
Vaccine Regimen: 6MHP (yes/no)	12MP+tet vs. 12MP+6MHP	0.0168*	2.2081	1.1533 to 4.2275
Age	≤ 40 vs > 40 years	0.0267*	0.1053	0.0144 to 0.7712
ECOG PS score	1 vs 0	0.0435*	2.3358	1.0250 to 5.3231
Advanced disease status	Recurrence vs initial diagnosis	0.0619	0.5439	0.2870 to 1.0308

When the model is run with study arms as a covariate, and stepwise selection is performed, a similar model is generated, now including study arm (D vs A):

Refined model (p = 0.0001, Chi-squared = 23.187): OS for 2.5y landmark (n = 138) with study Arm				
Covariate	Detail	p value	HR	95% CI
Study Arm	D (12MP+6MHP+Cy) vs. A (12MP+Tet)	0.0185*	0.3429	0.1407 to 0.8356
Age	≤ 40 vs > 40 years	0.0250*	0.1025	0.0140 to 0.7509
ECOG PS score	1 vs 0	0.0144*	2.8553	1.2326 to 6.6143
Advanced disease status	Recurrence vs initial diagnosis	0.0737	0.5608	1.2326 to 1.0571

Thus, the multivariable analysis of the 2.5y landmark data supports the significance of both vaccine regimen and study arm, when controlling for other covariates. We have added this information to the manuscript. These latter two models are included now in a modified **Table 3**, and the Results explains the findings with a modified paragraph, shown here:

Among participants evaluable after the 2.5 year landmark (n = 138), Cox regression analysis identified significant associations with OS for study arm (A vs D, p = 0.019), age and ECOG PS, if study arm was input in the model (**Table 3**). If vaccine regimen and Cy arms were input instead of arm, the refined model included vaccine regimen (p = 0.017), age and ECOG PS; **Table 3**). Thus, in both univariate and multivariable analyses, cognate CD4 help (6MHP) was a significant driver of OS.

As pointed out by the reviewer, sex is not a significant covariate in the multivariable analysis of the 2.5 year landmark data for OS. We have added a new supplemental table 4 that shows the OS for this subset by vaccine regimen and sex, and have added the following text as the latter half of the same paragraph above, in Results:

Sex was not a significant covariate for OS in the Cox regression analysis for this 2.5 year landmark dataset, but the HR for OS by vaccine regimen was more favorable for males (0.49; 95% CI 0.23-1.04) than for females (0.62; 95% CI 0.20-1.94), favoring the 6MHP regimen. Also, OS estimates at 10 years (7.5 years from the landmark) were 0.84±0.05 (SE) and 0.72±0.11 for males and females, respectively, receiving 6MHP vaccines and 0.70±0.07 and 0.63±0.11 for those receiving tet vaccines (**Supplemental Figure 4**).

In the Discussion (3rd paragraph from the last), we added the following:

In the 2.5 landmark analysis, sex was not a significant covariate for OS, but even in that dataset, males who received the 6MHP vaccines had more favorable survival, and the HR for benefit of the 6MHP vaccines was more favorable for males than females.

For RFS, the 2.5 year landmark requires excluding all for whom RFS was < 2.5 years, and this leaves only 83 participants, and no model with all the variables has significance overall. Instead, we developed a multivariable model for RFS at a 1 year landmark (n = 114). The model was created stepwise, removing if p > 0.5 and keeping if p < 0.1. In this model, sex was significantly associated with RFS (p = 0.023), with longer RFS for males (HR 2.094 for females vs males). However, no other covariates were significant. This analysis is limited by the small number of patients evaluable.

Optimized model (p = 0.0255, Chi-squared 4.992): 1 year Landmark for RFS (n = 114)

Covariate	Detail	p value	HR	95% CI
Sex	Female vs male	0.0230*	2.0940	1.1072 to 3.9605

This has also been summarized in the Results in the revised manuscript, in the following paragraph:

Models for RFS were also assessed in landmark analyses. Only 83 participants were evaluable for RFS after 2.5 years, and no model with all the variables has significance overall. Instead, we developed a multivariable model for RFS at a 1 year landmark (n = 114), where sex was significantly associated with RFS (p = 0.023), with HR 2.094 for females vs males (data not shown). However, no other covariates were significant.

Comment #2. It is unusual to observe survival benefit in stage II, III patients but not in stage IV patients. It is worth checking the clinical features associated with stage IV and address in a multivariable model whether the unusual survival benefit from stage can be explained by the concept of co-linearity.

Response.

Our multivariable analysis above (in response to question #1 above) does identify significant association (HR 0.5187, 95% CI: 0.2710-0.9925, p=0.0474, stage IV vs II-III) between stage and

OS for the full data set, adjusting for age, sex, 6MHP (yes/no) and Cyclophosphamide (yes/no). Further, we performed the following subgroup multivariable analyses for stage II-III patients and stage IV patients, respectively. First, we have run Cox regression analysis for multivariable analysis for OS for the subset of 132 patients with stage II-III (AJCC v8) disease. Incorporating the covariates for vaccine regimen (6MHP yes/no) and Cy arms (yes/no), plus the 5 clinical covariates (excluding stage since grouped by stage), and performing stepwise selection, the following model was generated, where there was significant association with favorable OS for the 6MHP vaccine regimen, and for age.

Refined model (p < 0.0001, Chi-squared = 24.418): OS for stage II-III (n = 132)				
Covariate	Detail	p value	HR	95% CI
Vaccine Regimen: 6MHP (yes/no)	12MP+tet vs. 12MP+6MHP	0.0347*	1.7750	1.0421 to 3.0235
Age	≤ 40 vs > 40 years	0.0067*	0.0650	0.0090 to 0.4695

We also performed a multivariable analysis for RFS including the same 7 covariates as for OS, and after stepwise refinement, a significant model was generated, and we found that the only significant covariate was vaccine regimen (p = 0.0447), with age approaching significance, as shown below:

Refined model (p = 0.0187, Chi-squared = 7.954): RFS for stage II-III (n = 132)				
Covariate	Detail	p value	HR	95% CI
Vaccine Regimen: 6MHP (yes/no)	12MP+tet vs. 12MP+6MHP	0.0447*	1.6355	1.0116 to 2.6441
Age	≤ 40 vs > 40 years	0.0673*	0.5020	0.2399 to 1.0503

Second, we tried to create a regression model for **stage IV** patients, but they represent only 35 patients; so, there were too few to include all the variables in a model. When we included just sex and vaccine regimen (6MHP yes/no), the following models were generated for OS and RFS, respectively:

Focused model (p = 0.7891, Chi-squared = 0.474): OS for stage IV (n = 35)				
Covariate	Detail	p value	HR	95% CI
Vaccine Regimen: 6MHP (yes/no)	12MP+tet vs. 12MP+6MHP	0.4896	1.5289	0.4585 to 5.0980
Sex	Female vs male	0.9686	0.9761	0.2929 to 3.2530

Focused model (p = 0.2739, Chi-squared = 2.590): RFS for stage IV (n = 35)				
Covariate	Detail	p value	HR	95% CI
Vaccine Regimen: 6MHP (yes/no)	12MP+tet vs. 12MP+6MHP	0.2382	1.7070	0.7020 to 4.1509
Sex	Female vs male	0.4016	1.4578	0.6042 to 3.5178

Thus, for stage II-III patients, the vaccine regimen is significantly associated with OS and RFS, with age significant for OS and approaching significance for RFS. This is now explained in the Results in the next-to-last paragraph of the Results section, with the model in new Supplemental Table 10:

To understand factors that may explain findings in univariate analyses within stage II-III participants, we performed multivariable analysis including the same covariates as for OS except stage, and after stepwise refinement, the only significant covariate was vaccine regimen (p = 0.045), with Age approaching significance (**Supplemental Table 10**).

There were only 35 patients with stage IV melanoma; so, there were too few to include all the variables in a model.

Comment #3. The manuscript highlighted the comparison of arm A and D, hypothesizing accordingly that Cy should be administered with 6MHP. However, there has been criticism from previous reviewers challenging the reliability of this hypothesis. The 2*2 design assumes independent / additive treatment benefit from 6MHP and Cy. An exploratory multivariable analysis can therefore be carried out to strengthen the hypothesis. I recommend that the treatment arm be represented using two binary variables 6MHP yes/no and Cy yes/no, so the survival benefit from each treatment can be quantified.

Response.

See the information in response to Comment #1 above. We hope this addresses the concern adequately.

Minor comments:

1. Label the treatment arms in a consistent way across all figures, e.g., arm a,b,c,d vs. 12MP + tet.

Response: We appreciate the guidance, but we are concerned that it will be harder for readers to follow if the figures are only labeled with study arm (A-D) in each case. If this is a strong consensus from the reviewers, we will certainly reconsider the request, but we hope this is acceptable.

2. Label the figure with p-values even if it is not significant and therefore not quoted in the main text.

Response: We have added p values and/or hazard ratios for all the Kaplan-Meier plots in Figures 2, 3, and 4 and in Supplemental Figures 1 and 4. We also have revised slightly Figure 4D, as the figure legend overlapped a curve in the original: that has been corrected. Hazard ratios are provided for Supplemental Figures 2, 3 and 5.

3. Change “multivariate analysis” into “multivariable analysis”. The former refers to analysing multiple dependent variables in statistics.

Response: *These changes have been made throughout.*

REVIEWERS' COMMENTS

Reviewer #4 (Remarks to the Author):

I am delighted to go through the additional analysis that the authors have made. The comments have been properly address and I believe the revised manuscript provides convincing evidence supporting the conclusion.

Point-by-point reply to reviewer comments:

“Multipeptide vaccines for melanoma in the adjuvant setting: long-term survival outcomes and exploratory analysis of a randomized phase II trial.”

We thank the reviewers for the favorable review from and positive comments. We have replied in detail to the prior reviews.

Reviewer #4 stated: “I am delighted to go through the additional analysis that the authors have made. The comments have been properly address and I believe the revised manuscript provides convincing evidence supporting the conclusion.”

We appreciate the favorable review.